# Formulation and Processing Strategies to Reduce Acrylamide in Thermally Processed Cereal-Based Foods

**DOI:** 10.3390/ijerph20136272

**Published:** 2023-07-01

**Authors:** Cennet Pelin Boyaci Gunduz

**Affiliations:** Department of Food Engineering, Faculty of Engineering, Adana Alparslan Turkes Science and Technology University, 01250 Adana, Turkey; cpbgunduz@atu.edu.tr

**Keywords:** contaminant, nutrition, thermal process, public health, toxicology, mitigation, exposure, risk, health, dietary intake, food safety

## Abstract

Acrylamide, a thermal process contaminant, is generated in carbohydrate-rich foods processed at high temperatures (above 120 °C). Since acrylamide indicates a human health concern, the acrylamide contents of various foods and the dietary exposure of the population to acrylamide are very well investigated. Commonly consumed foods in the daily diet of individuals such as bakery products, potato products and coffee are major dietary sources of acrylamide. In recent years, dietary exposure levels of the population and mitigation measures for reducing acrylamide in different food products have gained importance to decrease the public’s exposure to acrylamide. Since the complete elimination of acrylamide in foods is not possible, various mitigation measures to reduce acrylamide to levels as low as reasonably achievable have been developed and applied in the food industry. Mitigation strategies should be applied according to the different product categories during agricultural production, formulation, processing and final consumer preparation stages. The aim of this review is to evaluate formulation and processing strategies to reduce acrylamide in various cereal-based food products and to discuss the applicability of mitigation measures in the food industry by taking into consideration the organoleptic properties, nutritional value, cost and regulations in the light of current knowledge.

## 1. Introduction

Food processing is the use of various methods and techniques to convert agricultural products such as grains, vegetables, fruits, meats and milk into food ingredients or finished food products [1]. Thermal processing is one of the commonly applied food processing operations to reduce or destroy microbial and enzymatic activities, prolong the shelf life and improve the nutritional, sensorial and functional properties of the foods. On the other hand, thermal processing can lead to the formation of thermal process contaminants. Due to the adverse toxicological effects of the thermal process contaminants, they can pose potential health risks to individuals.

Various thermal processing contaminants have been identified in foods that require attention due to potential health concerns [2]. Acrylamide is one of the well-known thermal process contaminants that is generated in carbohydrate-rich foods processed at high temperatures. Especially, potato products, such as French fries and potato crisps; coffee and cereal products, such as bread, breakfast cereals, crackers, rusks, biscuits and baby foods contain acrylamide at various levels [3]. These foods are commonly consumed by people and, thus, acrylamide poses serious public health risks and concerns.

Acrylamide is classified as a Group 2A—probable human carcinogen by the International Agency for Research on Cancer [4]. Acrylamide has been extensively studied by food authorities and researchers for characterization of the risk-by-risk assessment and toxicological studies, investigation of possible generation pathways in food products, determination of acrylamide levels of different food products, calculation of exposure levels for the various population and age groups and, also, development of mitigation strategies to decrease acrylamide in foods.

Exposure to acrylamide is affected by the acrylamide concentration in the food product, the amount of food consumed by an individual, and body weight. Since acrylamide is found in commonly consumed foods, the exposure of the population could be moderate to high levels. Especially, babies and children are the most sensitive population to acrylamide as a result of their lower body weight and, also, their vulnerability to toxic compounds [5]. In that context, acrylamide levels in food products and the resulting acrylamide exposure of the population should be reduced. Therefore, the development and application of mitigation strategies are very important in the food industry. In addition, exposure assessments should be monitored regularly to evaluate the effectiveness of the mitigation strategies [6].

The aim of this review is to evaluate formulation and processing strategies to reduce acrylamide in various cereal-based food products and to discuss the applicability of mitigation measures in the food industry by taking into consideration the organoleptic properties, nutritional value, cost and regulations in the light of current knowledge.

## 2. Processing and Formulation Strategies to Mitigate Acrylamide in Different Thermally Processed Cereal-Based Food Products

Cereal-based products are the main component of the daily diet. Commonly consumed cereal-based foods including bread, biscuits, crackers, cookies, rusks, cereal-based baby foods and breakfast cereals contain different levels in the range of 38–407 μg/kg at average and 60–1600 μg/kg at the 95th percentile [3]. Table 1 shows mean and 95th percentile (P95) acrylamide levels according to the EFSA report (2015) and benchmark levels set by the European Union, Commission Regulation (EU) 2017/2158 [7]. Many survey studies were conducted on acrylamide levels in cereal-based food products worldwide. Depending on the applied technology, processing conditions, raw materials and national production methods and regulations, acrylamide levels can change in the products worldwide.

Acrylamide poses serious health concerns and individuals can be exposed to acrylamide through the consumption of acrylamide-rich foods. Cereal-based products are the source of acrylamide exposure due to their high consumption. Therefore, it is very important to decrease acrylamide levels in cereal-based products to reduce dietary acrylamide exposure. In that context, mitigation strategies and, also, regulatory guides were developed to decrease acrylamide levels as low as reasonably achievable in different food product categories to reduce the acrylamide exposure of the population. The European Union established mitigation measures and set benchmark levels for decreasing the acrylamide levels in foods [7]. Figure 1 shows the benchmark levels for each cereal-based food product categorized in different groups according to the Regulation set by European Union, Commission Regulation (EU) 2017/2158. The FoodDrinkEurope (FDE) developed “Acrylamide Toolbox” as a guide with possible intervention steps to prevent or reduce acrylamide below the benchmark levels in different food categories [8]. The Food and Drug Administration (FDA) published guidance for acrylamide mitigation in the food industry [9].

Acrylamide content varies in cereal-based products according to the amounts of precursors (free asparagine, glucose, fructose, and maltose), the food matrix and the type and extent of the thermal process applied [10]. The amino acid asparagine is the main precursor of acrylamide. It reacts rapidly with carbonyls and reducing sugars via the Maillard reaction and results in acrylamide [2,11,12]. Since free asparagine is the main precursor of arylamide, raw materials low in asparagine could have a lower acrylamide potential [8].

Acrylamide levels should be as low as reasonably achievable (ALARA) in the foods to prevent acrylamide exposure of the population from a public health point of view. There is not a safe lowest label, but the benchmark levels may provide reference values for the food industry [13]. In that context, different science-based mitigation techniques were developed during various stages of food production and consumption to achieve acrylamide levels as low as reasonably achievable below benchmark levels [8]. On the other hand, it is very important not to affect the existing nutritional value, chemical and microbiological safety, organoleptic characteristics such as texture, taste, color, and shelf life of the final product during the application of mitigation measures [5]. In that context, different strategies should be developed depending on the food category and mitigation strategies should be applied according to the product category at different stages of the food product chain. Mitigation measures should be applied based on different parameters during different stages (Figure 2) according to the product category as reported by the FoodDrinkEurope Acrylamide Toolbox [8]. It is significantly important to apply strategies depending on the product type and stage without affecting the final nutritional value, organoleptic characteristics and safety, but also, compatibility in the food industry, costs and regulatory compliance should be well evaluated for each strategy.

### 2.1. Formulation and Processing Strategies in Cereal-Based Products

Acrylamide levels in cereal-based foods vary depending on the acrylamide precursors, mainly free asparagine and also reducing sugars, food matrix and thermal processing conditions. At the formulation and product design stage, using different types or parts of grain with low free asparagine content, fully or partly replacing NH_4_HCO_3_ with alternative raising agents, the use of asparaginase, re-evaluation of usage of heat-treated co-ingredients, replacing fructose or fructose-containing ingredients, addition of sugar-based co-ingredients after heat treatment stage, certain amino acids addition, organic acid addition, consider piece size/surface area to volume ratio during product design are among the recommended applications depending on the product to reduce acrylamide (Figure 3). During the processing of the cereal products, extension of fermentation time, optimization of baking temperature, temperature profile, time and moisture, avoiding burnt products and production of lighter color final products are mitigation strategies applied in different cereal-based products [8]. For example, in a recent study, it was reported that acrylamide levels in breads decreased with the extension of fermentation time, and especially when fermentation reached 10–12 h [14]. Another study concluded that decreasing the baking temperature of biscuits from 200 °C to 180 °C in the same conditions (pH, baking time, and sugar concentrations) decreased acrylamide levels in the final product more than 50.00% [15]. End product color in biscuits was evaluated with some other characteristics in a study and lower water activity and higher color indices resulted in higher acrylamide levels in biscuits [16].

In that context, mitigation strategies should be well evaluated according to the product category to improve food safety, quality and sensorial attributes of the final product.

#### 2.1.1. Bread

Acrylamide levels vary in bread depending on the time and thermal processing, acrylamide precursors found in different grains, the grain varieties and parts of the grains, fermentation time and type of microorganisms and used co-ingredients. Free asparagine in combination with reducing sugars processed at high temperatures (>120 °C) generates acrylamide [11,12] and in cereal grains, free asparagine concentration is the major determinant of acrylamide formation [17,18,19,20]. Free asparagine content changes in different species of cereals. Zilic and others (2017) reported the average free asparagine contents of various cereal species between 426 ± 144 and 1179 ± 359 mg/kg [21]. The rye and oat contain higher free asparagine than that of wheat flour and, therefore, they have an important potential for acrylamide formation [21,22,23]. In a previous study conducted on acrylamide levels in bread samples, higher acrylamide contents were determined in rye bread compared to wheat bread types [24]. Free asparagine in the grains is not the only factor resulting in acrylamide generation, but, has a significant importance as the limiting precursor in the formation of acrylamide in cereal-based baked foods [25]. Therefore, the free asparagine content effects the acrylamide level in the final product. In a study, free asparagine was higher in the whole rye fermented dough (7585 ± 158 mg/kg) and acrylamide was determined as the highest in the whole rye bread crust (1565–3887 µg/kg) prepared from rye flour in the bread crust-like model systems compared to other cereals [26]. In addition, parts of the grains used in the formulation affect the acrylamide content since the free asparagine content is more concentrated in the germ and bran parts [8]. In that context, using grain types with lower asparagine contents and lowering bran and germ contents in bread formulations may result in lower acrylamide levels. However, those kinds of modifications will change the bread type and final organoleptic and nutritional properties which are important in consumer preferences and choices. Because consumers prefer certain bread types based on sensorial and nutritional properties. In that context, it is not easy to completely change the bread formulation since it will alter the bread type. Complete elimination of bran and germ would not be possible, since some breads with high bran content are preferred by consumers due to fiber content and some other nutritional compounds.

In addition, reducing the flour extraction rate can decrease asparagine content in the flour, but lower flour yield could result in economic losses [25]. Asparaginase applications or assessment of the other co-ingredients could be other strategies applied during the formulation stage. For example, addition of other flour types, grains and grain parts or different nuts or dried fruits additions could increase acrylamide levels. Fortified bread types with nuts and seeds can contain higher acrylamide due to the added co-ingredients. Because, added co-ingredients could contain higher amounts of acrylamide precursors. In addition, some kinds of nuts and seeds are roasted before addition to the formulation and could result in higher acrylamide levels in the final product. Therefore, the impact of those kinds of co-ingredients that have acrylamide levels that raises the potential in the final product should be considered during the formulation stage. For example, the use of nuts and seeds roasted at lower rather than higher temperatures or replacement of nuts and seeds containing a higher amount of fructose with glucose [7,8].

Processing conditions including baking temperature, temperature profile and time should be well-optimized to enable lower thermal inputs. In a recent study, decreasing the baking temperature from 260 °C to 230 °C reduced the mean acrylamide content in the final rye bread samples by ∼25% [27]. Baking bread to a lighter color endpoint could be another mitigation strategy applied during the thermal processing step in breads [7]. The bread crust color could be an indication of acrylamide. Most of the generated acrylamide is found in the crust. Crumb contains trace levels of acrylamide as a result of the distribution of the thermal load and moisture maintenance [10]. Significant correlation between crust color and acrylamide content until a certain point was reported previously [28,29]. In that context, dark bread, crust-colored and burnt products should be prevented.

During the processing stage, increasing the fermentation time of the dough is an effective way to decrease acrylamide levels, which is mainly related to the ability of yeasts to metabolize acrylamide precursors such as asparagine [8]. In addition, the fermentation process conducted with yeasts and also, lactic acid bacteria can reduce acrylamide content in bread at higher ratios related to a decrease in pH [30]. Sourdough bread is produced with a mixture of flour and water that is fermented by lactic acid bacteria and yeasts [31] and the use of sourdough in bread production improves the functional properties of the dough and final bread due to the metabolites that are produced during sourdough fermentation as a result of the activities of the microbial flora [32]. Breads fermented by appropriate lactic acid bacteria strains together with yeast can decrease acrylamide levels compared to bread fermented only by yeast [33]. Lactic acid bacteria produce lactic acid from the fermentation of carbohydrates as the main metabolite and the pH in sourdough bread decreases to below 5 [34]. It was reported that acrylamide levels in sourdough bread fermented with lactic acid bacteria and yeasts were less than that in yeast-fermented bread which is mainly related to glucose metabolism and pH drop associated with acid production [35]. In that context, fermentation with suitable lactic acid bacteria strains and the yeast *Saccharomyces cerevisiae* can be used to reduce acrylamide levels in bread [36]. In a study, acrylamide content in wheat breads fermented with different combinations of lactic acid bacteria cultures, *Pediococcus pentosaceus* and *Limosilactobacillus fermentum* (formerly *Lactobacillus fermentum*), and *Saccharomyces cerevisiae* were investigated and it was reported that sourdough fermentation with appropriate strains can be used as a mitigation technology to reduce the acrylamide content of wheat breads since acrylamide content reduced 24.38%–58.83% by sourdough fermentation as a result of the combined effects of the increased levels of sourdough acidity and the reduced levels of precursors due to the consumption by micoorganisms [14]. The phenomenon for acrylamide reduction can be strain dependent due to the limiting factor of the reactants or decreased pH [37]. Lactic acid bacteria excreting lower amylolytic activity and higher proteolytic activity can be effective to reduce acrylamide in breads [38]. Lactic acid bacteria strains investigated to reduce acrylamide in bread in previous studies include *Pediococcus acidilactici*, *Pediococcus pentosaceus*, *Lactiplantibacillus plantarum* (formerly *Lactobacillus plantarum*), *Latilactobacillus curvatus* (formerly *Lactobacillus curvatus*), *Latilactobacillus sakei* (formerly *Lactobacillus sakei)*, *Lacticaseibacillus paracasei* (formerly *Lactobacillus paracasei*), *Levilactobacillus brevis* (formerly *Lactobacillus brevis*), *Leuconostoc mesenteroides, Limosilactobacillus reuteri* (formerly *Lactobacillus reuteri*), *Lacticaseibacillus casei* (formerly *Lactobacillus casei*), *Lacticaseibacillus rhamnosus* (formerly *Lactobacillus rhamnosus*), *Lactobacillus delbrueckii*, *Limosilactobacillus fermentum* (formerly *Lactobacillus fermentum*) [14,38,39,40,41,42,43,44]. It was reported that application of specific strains of probiotics especially *Lactobacillus* spp. can be used as a mitigation approach to reduce acrylamide. The research on probiotics and acrylamide reduction is limited, but as stated previously, the main mechanism to reduce acrylamide could be the binding of peptidoglycan components in the bacteria cell wall to acrylamide or the production of the asparaginase enzyme by specific strains [45,46]. From a cost-effective point of view, lactic acid bacteria cultures are expensive. In addition, sourdough fermentation needs more attention and time for back-slopping to hold the cultures in the active state. Moreover, some types of sourdough bread production need more fermentation time. In the food industry, efficient and short time production is important from an economical point of view. Sourdough bread is produced more at artisanal level, but, also at industrial scale, too. For industrial production, breads produced with lactic acid bacteria and yeast culture combinations could be an effective mitigation strategy. In that context, strains having the capability to produce less acrylamide in the final breads could be a technological characteristic that can be investigated for designing starter culture combinations at the industrial level.

#### 2.1.2. Other Bakery Products—Biscuits, Crackers, Cookies, Wafers and Rusks

Acrylamide levels show a wide variation in the products of that category since each product category and the different products in each category have different time and temperature norms during processing, final moisture levels and diverse formulations including grain types and parts, ingredients, baking agents and, also, various innovative and functional properties which affect the final product. In that context, depending on the product, various mitigation strategies could be applied.

As written in the previous section, using grains low in asparagine contents and using less bran and germ contents in the formulations may result in lower acrylamide levels. Different parts of the grains used in the formulations show higher acrylamide levels, such as mainly bran-based biscuits contain higher acrylamide compared to plain-wheat biscuits [47]. A recent study showed that cake made with whole-meal flour contained more acrylamide than cake prepared in white flour in the formulation [48]. It was previously reported that the highest levels of acrylamide were detected in oat, rye and teff biscuits among all biscuit types [47,49]. On the other hand, lower acrylamide formation was reported in corn and rice-based biscuits [49]. Re-formulation of biscuits using grains and grain parts that contain lower free asparagine would be a strong mitigation strategy to reduce acrylamide in biscuits. However, those kinds of modifications will completely change the type and affect the final organoleptic and nutritional properties. In that context, recipes and designs in formulations can assess partial replacement.

Another strategy that may be applied in bakery products is reducing or replacing fully or partially ammonium bicarbonate with alternative raising agents such as sodium bicarbonate [7,8]. However, this can increase overall sodium content from a nutritional point of view and, also, result in organoleptic changes which will affect product identity and consumer acceptance [7]. Some products such as sweet biscuits and gingerbread have been reformulated and commercialized despite changes to flavor, color and texture. Combinations of ammonium bicarbonate with alternative raising agents such as sodium bicarbonate and acidulants are often required to affect the properties of the final product to a lesser extent [8]. In a previous study, it was reported that ammonium bicarbonate was the most effective ingredient to reduce acrylamide in cookie formulations [50].

Asparaginase can be used in formulations of bakery products to reduce acrylamide [7]. In recent years, microbial L-asparaginase produced from microorganisms has been gaining importance, with potential applications in the pharmaceutical and food industries [51,52]. The enzyme asparaginase catalyzes the hydrolysis of L-asparagine into aspartic acid and ammonia and if it is used in the formulations, acrylamide formation is significantly inhibited due to the conversion of the main precursor of acrylamide, asparagine, into aspartic acid [53]. For the food industry, commercial food-grade enzymes, Acrylaway® (Novozymes, Bagsværd, Denmark) and PreventASe® (DSM, Heerlen, The Netherlands) obtained from fungi, *Aspergillus oryzae* and *Aspergillus niger*, have been developed for the prevention of acrylamide formation in foods including cookies, biscuits, crackers and crisp and toasted breads, with no effect on the organoleptic properties of the final products [54]. According to the Joint FAO/WHO Expert Committee on Food Additives (JECFA), asparaginase from *A. oryzae* does not represent a hazard to human health to be used in the formulations before the thermal processing stage to prevent acrylamide formation. The limitations of this strategy include that there is limited or no effect in recipes with high-fat content, low moisture or high pH value [8]. The main formulations and conditions can affect enzyme activity. In addition, national and international regulations should be taken into consideration for using asparaginase enzyme in cereal dough formulations. From a cost-effective point of view, enzyme development and application in the industry could be expensive. The development of cost-effective asparaginase enzyme with desirable properties will reduce the economic cost of its application in the food industry [52]. In the study of Rottram and others (2021), asparaginase treatment reduced the acrylamide level by 59% without affecting color or taste and it was reported that use of a 0.5% enzyme solution resulted in a reasonable cost/effect-ratio. As it was concluded in the study, the enzyme treatment is costly and depending on the activities used, the additional costs of 0.05 (for 0.1% enzyme) to 0.5 USD/kg (for 1.0% enzyme) can be estimated [55]. However, this study was conducted in French fries. In cereal-based products, enzyme treatments cost could be different due to the different concentration of enzyme solutions that are needed to reduce acrylamide effectively. Anese and others (2011) tested different formulations of biscuits and reported differences in the efficiency of L-asparaginase. As it was reported, this could be related to the different compositions of the products and L-asparaginase showed a better reduction of acrylamide in biscuits with high water content [56]. In that context, costs could be changed according to the applied enzyme concentration depending on the formulation.

Different kinds of ingredients are added to dough formulations, especially biscuits, to develop new products. In that context, biscuits can contain various co-ingredients such as nuts, seeds and dried fruits which can contribute to acrylamide generation. Therefore, those kinds of ingredients can raise acrylamide levels in the final product. Depending on the composition, the roasting time and temperature of those kinds of co-ingredients cause acrylamide levels to change in the final product [57]. Acrylamide levels were reported higher in cocoa-based biscuits than wholegrain and wheat biscuits in the previous studies [15,47]. Higher amounts of acrylamide in cocoa-based biscuits could be related to the additional acrylamide load related to cocoa beans roasting [15]. Since the cocoa beans roasting temperature will affect the acrylamide content in the final product, the processing conditions during the roasting stage should be evaluated. As a result, the composition and thermal processing of the added co-ingredients should be well evaluated. For example, almonds roasted at lower temperatures rather than higher temperatures can be preferred. Because almonds are susceptible to acrylamide formation when the almond kernel temperature is >130 °C since they include high levels of asparagine and sugars. In a previous study, acrylamide levels increased in the almond samples from 235 to 907 µg/kg as the roasting temperature increased from 129 to 182 °C, respectively. The most significant increase in acrylamide was observed with the temperature increase between 146 and 154 °C, as the mean level of acrylamide almost doubled (from 323 to 642 μg/kg) [58].

As another re-formulation strategy, if the product design allows, replacing fructose or fructose-containing ingredients (e.g., syrups, honey) with glucose or non-reducing sugars (e.g., sucrose) could be an effective tool to reduce the acrylamide formation, especially in recipes containing ammonium bicarbonate [8]. However, replacing fructose or other reducing sugars may result in a modified product identity due to color and flavor changes.

Some other possible mitigation strategies that may be applied during the formulation stage depending on the product type include adding minor ingredients such as certain amino acids, divalent cations and organic acids. However, these minor ingredients may negatively affect the appearance, taste and texture of the final product. For example, organic acids addition can cause sour taste end product. The addition of organic acids/pH adjustment could decrease acrylamide levels by affecting the Maillard reaction, but result in organoleptic changes (such as less browning, modification of taste) [8]. Amino acids can be used as a mitigation strategy due to the mechanism of competing for available Maillard reaction intermediates or reacting with acrylamide itself through Michael addition. In previous model studies, lysine, cysteine, γ-Aminobutyric acid (GABA) showed high capacity to reduce acrylamide content in thermal processed foods [59,60]. However, the addition of those kinds of ingredients should be well evaluated at each product category and type since those kinds of modifications in the formulations affect the sensorial properties of the final product. For example, glycine addition reduced acrylamide content during the pilot scale trials in bakery products such as sweet biscuits, gingerbread and crisp bread, but affected the sensorial properties and quality of the final products undesirably [8]. A recent study investigating the effects of sodium alginate, chitosan and pectin on the acrylamide formation of air fried biscuits reported that the biscuits with hydrocolloids addition had lower acrylamide than control biscuits and the texture, flavor and overall acceptability of biscuits produced with hydrocolloids addition were not significantly different than control, except for the odor and appearance [61]. Other minor ingredients, calcium salts or organic acids can be added to formulations to reduce acrylamide concentrations through different pathways in bakery products. However, the application of these ingredients is limited at the commercial plant level due to the resulting organoleptic changes and affecting key quality attributes negatively [8].

In addition, the shape and size of biscuits, crackers and crisp breads affect the acrylamide levels in the final product. Due to the heat impact, the smaller piece of size of the product potentially leads to higher acrylamide levels [7,8]. Acrylamide levels could be affected according to the size and shape of the cross-section of the product (surface area to volume ratio) [62]. In a study conducted in potato-based products, at the same frying temperature and time conditions, the slice size of potato varieties affected the final acrylamide concentration [63]. The study examined the acrylamide concentrations of two different varieties at three different slice size, temperature and time intervals and concluded that product size effects the thermal process as a result of heat transfer. In that context, acrylamide content can decrease by increasing slice size at the same temperature and time conditions. In the study, approximately 20–24% and 44–48% fold increase in acrylamide content was observed at 150 °C when the slice size decreased from 9 mm to 6 mm and to 3 mm, respectively. At 170 °C and 190 °C approximately 12–15% and 24–28% fold increase in acrylamide content were observed when the slice size decreased from 9 mm to 6 mm and to 3 mm, respectively [63]. However, this modification cannot be applied to all products since it significantly changes the product appearance and characteristics.

During the processing stage, thermal input and final product moisture are important parameters to follow to prevent high amounts of acrylamide formation. In that context, processing conditions, baking temperature, temperature profile and time should be well-optimized to enable lower thermal input. Products with crisp textures that are baked at a high temperature and to a low final moisture content tend to be higher in acrylamide [8]. A recent study investigated the influence of temperature on the formation of acrylamide in bakery products and acrylamide levels were higher in the biscuits, muffins and cakes baked at 190 °C than 170 °C. The acrylamide levels in cakes, muffins and biscuits baked at 170 °C and 190 °C were as follows: 71.21 μg/kg and 81.19 μg/kg, 84.24 μg/kg and 102.84 μg/kg and 126.52 μg/kg and 151.52 μg/kg [64].

Heat transfer mode could affect the acrylamide formation. In a recent study, static and ventilated baking modes on biscuits were investigated and at 175 °C for moderate baking times (20–22 min), the ventilated mode resulted in higher acrylamide levels [65].

In addition, dark-colored and burnt products should be prevented in the products of that category. It was reported that the browning index calculated based on color measurements in biscuits can be used as a screening tool since high browning index was associated with high acrylamide content [66]. As reported in the previous section, increasing the fermentation time of the dough is an effective way to decrease acrylamide levels during the processing stage.

#### 2.1.3. Breakfast Cereals

Breakfast cereals have different formulations depending on the type and target consumers. Different grain types are used in breakfast cereals formulations according to the consumer preferences. Since certain grains have less potential to produce acrylamide, such as maize and rice, compared to wheat, rye, oats and barley, maize and rice can be used in new product formulations. In addition, using less whole meal or bran and more endosperms could result in final products containing less acrylamide. However, these kinds of modifications may change partly or completely the product identity, sensorial properties and nutritional quality [8]. Because reducing the whole-meal part in the formulations can affect the contents of fiber and other beneficial nutrients which are important in consumer preferences.

Depending on the type of applied process, breakfast cereals can include honey or some co-ingredients such as dried fruits and nuts which can lead to higher acrylamide levels in the final product. Therefore, the ingredients containing reducing sugars such as honey should be added after heat-treatment stages to prevent them from acting as precursors to acrylamide formation [7]. Added co-ingredients such as roasted nuts should be well evaluated not to result in final products with acrylamide contents above benchmark levels. Because the contribution to the acrylamide levels of each dried fruit and nut will be different depending on the composition and thermal processing.

Using the asparaginase enzyme has limited application in breakfast cereals since low moisture content during their processing makes it difficult for the enzyme to penetrate into the grain or food matrix [8]. As discussed in the previous parts, the addition of other minor ingredients, glycine as the amino acid, calcium salts or organic acids, or the reduction of phosphate salts in the formulations to reduce acrylamide levels can affect organoleptic properties such as color, flavor and texture negatively.

During the processing stage of the breakfast cereals, the combination of moisture content and thermal input should be optimized to reduce acrylamide. Burnt products should be prevented. Moisture content is important in acrylamide levels, since the low moisture content is related to higher thermal input, but high moisture can affect the shelf life. In that context, all parameters should be optimized taking into consideration the final product from different aspects.

#### 2.1.4. Cereal-Based Baby Foods

The acrylamide contents of baby foods show a great variation according to the type, formulation and processing method. Especially, cereal based baby foods are frequently consumed by infants and toddlers and due to the common consumption, they contribute considerably high acrylamide exposure [5]. In many comprehensive exposure assessment survey studies conducted in different continents, baby foods such as baby biscuits, crackers and rusks are an important contributor to acrylamide intake in lower age groups [3,67,68]. EFSA (2015) reported that processed cereal-based baby foods contributed 30% of the total exposure level and crackers, biscuits and crispbread contributed 20% of the total exposure [3]. Thus, for babies and children, acrylamide is considered a concern. Therefore, acrylamide levels should be as low as reasonably achievable below the benchmark levels. Lower limits were set as benchmark levels for baby foods, 150 µg/kg for infant biscuits and rusks and 40 µg/kg for processed cereal foods excluding biscuits and rusks, as shown in Figure 1.

Acrylamide contents of commonly consumed foods by babies and children should be a high safety concern. Mitigation strategies should be applied in the food products containing higher amounts of acrylamide than the benchmark levels to decrease the exposure of babies. Current recommended mitigation strategies to decrease acrylamide levels in baby foods include formulation strategies such as using raw materials with low acrylamide precursor content and asparaginase applications, and processing strategies as an effective combination of temperature and heating times [7].

During the formulation stage, grains and grain parts which tend to generate less acrylamide can be used. But, as reported in the previous parts, product identity, sensorial properties and nutritional value can be affected. Adding ingredients that may contribute to increasing reducing sugars should be avoided since reducing sugars added to the mix such as fruits, honey, and fructose leads to a higher amount of acrylamide in the final product [8]. As it was reported previously, the infant cereals including honey and grape juice concentrate could result in more acrylamide which shows the importance of formulation of baby foods [69]. As reported in the previous sections, asparaginase could be a very effective mitigation strategy in certain infant cereal processes and can be applied commercially since the production and processing stages of baby foods are suitable for enzyme applications. Because the recipes of most baby foods include a large proportion of cereal flour and water in the wet mix hydrolysis step which enables the use of asparaginase enzyme [8]. Therefore, depending on the type of cereal-based baby foods, the asparaginase enzyme can be applied.

As reported in the previous sections, thermal input and moisture content should be optimized during the processing stages. However, decreasing thermal input to reduce acrylamide formation should be very well evaluated to prevent microbiological risks. Therefore, it is considerably important not to affect chemical and microbiological safety and organoleptic characteristics such as taste, texture, color and, also, shelf-life of the products [7,8]. Therefore, each mitigation strategy should be well-evaluated based on the risk-benefit strategy to reduce acrylamide, but, at the same time, to protect the nutritional and sensorial properties of the final product.

## 3. Conclusions

Exposure to acrylamide is a public health concern and countries should develop regulations and benchmark levels to reduce exposure through consumption of acrylamide-rich foods. Acrylamide levels should be as low as reasonably achievable below the benchmark levels. In that context, effective mitigation strategies should be applied according to the product. There is not a direct or complete strategy for all products to completely eliminate acrylamide. In that context, all food product processing and formulations should be well evaluated individually to decrease acrylamide in the final product. Especially, long fermentation times, decreasing baking temperature, asparaginase treatment and production of lighter color final products could be effective mitigation strategies to apply in cereal-based products. Moreover, besides laboratory research experiments and pilot scale trials, commercial applications of the strategies should be discussed. Because a mitigation strategy that reduces arylamide at the pilot scale cannot be successful or cannot be applied commercially. In that context, mitigation strategies should be evaluated from all aspects such as compatibility with commercial applications in the food industry, cost-effectiveness and protecting nutritional value, microbiological and chemical safety. Moreover, organoleptic properties and product identity should not be changed since it is significantly important to modify products without reducing consumer acceptability.

## Figures and Tables

**Figure 1 ijerph-20-06272-f001:**
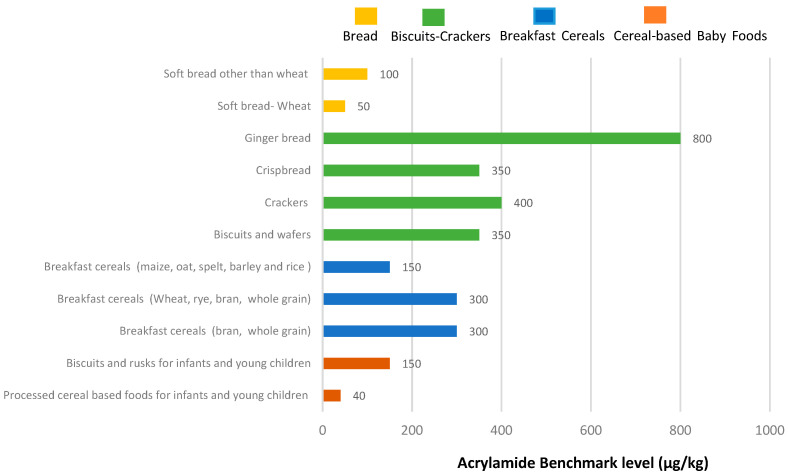
Benchmark levels for each cereal-based food product categorized in different groups according to the Regulation set by European Union, Commission Regulation (EU) 2017/2158.

**Figure 2 ijerph-20-06272-f002:**
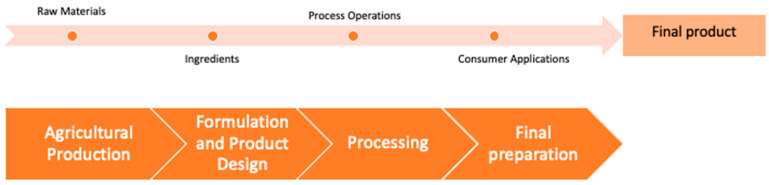
Different stages of the food product chain for application of mitigation strategies.

**Figure 3 ijerph-20-06272-f003:**
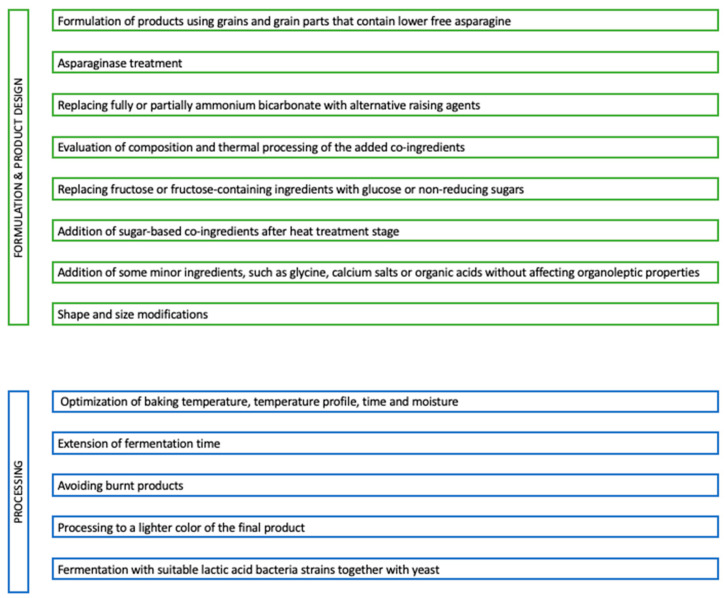
Possible mitigation strategies to reduce acrylamide in cereal-based food products [8].

**Table 1 ijerph-20-06272-t001:** Mean and 95th percentile (P95) acrylamide levels (µg/kg) according to the EFSA report (2015) and benchmark levels set by European Union, Commission Regulation (EU) 2017/2158 (2017).

Cereal-Based Product Categories	Acrylamide Levels (µg/kg)		
	Mean	P95	Benchmark Level
Soft bread-Wheat	38	120	50
Soft bread other than wheat	57	240	100
Breakfast cereals (bran, whole grain)	211	716	300
Breakfast cereals (Wheat, rye, bran, whole grain)	170	410	300
Breakfast cereals (maize, oat, spelt, barley and rice)	102	403	150
Biscuits and wafers	201	810	350
Crackers	231	590	400
Crispbread	171	486	350
Ginger bread	407	1600	800
Processed cereal based foods for infants and young children	89	60	40
Biscuits and rusks for infants and young children	111	287	150

## Data Availability

No new data were created or analyzed in this study. Data sharing is not applicable to this article.

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
