# Peer review of "Formulation and Processing Strategies to Reduce Acrylamide in Thermally Processed Cereal-Based Foods"

_ijerph, 2023, doi:10.3390/ijerph20136272_

Round 1
Reviewer 1 Report
Dear Authors, in my opinion the paper is inadequate to be published as a review paper for these main reasons:
the paper i organized in 5 main parts, the first one concerning general strategies to reduce acrylamide in cereal-based products and then 4 parts concerning specifically bread, other bakery products, breakfast cereals and baby foods, respectively. No tables and specific data are given for any of these product categories, one would expect some effective synthesis of information and numerical data through tables/diagrams. All of the above-mentioned paragraphs report the same statements and conclusions, mostly derived from references 7 and 8 (EFSA and Food and Drink Europe toolbox). Therefore there are many repetitions since these references are cited, alone or together, 27 times in the paper! I suggest to: a) rewrite the paper, maybe not dividing product categories since ingredients, critical points and strategies are mostly common; b) organize and report literature data, including acrylamide content/ranges in the variously treated food products, through tables/graphs, giving the appropriate references; c) write a unique conclusion instead of repeating the same suggestions many times.
Author Response
Dear Reviewer,
Thank you very much for your valuable comments to improve our paper. Corrections and additions were done in the text. Please find the point-by-point response below:
- Some parts are re-written as recommended. Same statements were updated with the new information throughout the text. New references were added.
- Mean acrylamide contents were presented in Figure 1. But, as you recommend, they are updated with ranges and converted to a Table to be more clear. In that context, Table 1 is added.
- Conclusion was improved.
Reviewer 2 Report
The manuscript is clear and of good value to the acrylamide content in food.
Please consider the following comments:
Abstract line 9: please specify the temperature (above 120...)
lines 129-132: It is recommended to mention values from the cited reference that would strengthen the point.
line 193: please mention if possible the change of pH.
line 279: By how much was the economical cost reduced?
line 293: please mention the difference between higher and lower temperatures numerically if possible since you are discussing factors to reduce acrylamide content so it is worth it whenever it is cited in literature to express results by numbers.
line 302 Can you please provide more details the difference in taste, and texture?
line 323: Please mention by how many folds did the acrylamide content increase.
line 332: By how many folds was the concentration of acrylamide higher at 190C?
Author Response
Dear Reviewer,
Thank you very much for your valuable comments to improve our paper. Corrections and additions were done in the text. Please find the point-by-point response below:
Abstract line 9: please specify the temperature (above 120...)
P1, L9: It is specified.
lines 129-132: It is recommended to mention values from the cited reference that would strengthen the point.
P4, L148-163 : Thank you. The values were added as you recommend to be more clear.
line 193: please mention if possible the change of pH.
P6, L253 : Thank you. The values were added as you recommend.
line 279: By how much was the economical cost reduced?
P8, L358-370: Economical assessment and cost of enzyme treatments were evaluated and added.
line 293: please mention the difference between higher and lower temperatures numerically if possible since you are discussing factors to reduce acrylamide content so it is worth it whenever it is cited in literature to express results by numbers.
P8, L402-405: Effects of roasting temperature was discussed and change in acrylamide amounts were added.
line 302 Can you please provide more details the difference in taste, and texture?
P8, L397-400: It was added.
line 323: Please mention by how many folds did the acrylamide content increase.
P8, L426-438: Changes according to the different slice sizes and fold increase in acylamide levels were discussed.
line 332: By how many folds was the concentration of acrylamide higher at 190C?
P10, L447-456: Concentration change was added.
Reviewer 3 Report
I have reviewed the review paper titled: Formulation and Processing Strategies to Reduce Acrylamide in 2 Thermally Processed Cereal-Based Foods
This review article aims to evaluate formulation and processing methologies to mitigate acrylamide formation in bakery products, breakfast cereals and cereal-based baby foods and to discuss the applicability of mitigation methods in the food industry by considering organoleptic properties, nutrition, cost and regulations in current status. The information of this work is useful and relevant and there are many strategies of the review paper that could be adapted by food processing industry especially for cereal based bakery, breakfast and baby foods in the future. I think the review paper is acceptable after minor revision. Although, the article is not innovative, it contains original researchs and interesting information to mitigate acrylamide formation among processing technologies and formulation for agriculture processing of foods. Abstract is well written upon and the thermal process contaminant, acrylamide and the applicability of mitigation methods and other consideration are mentioned. Introduction is well addressed including the formation of thermal processing contaminant, acrylamide, and its risk, common consuming food containing acrylamide and its concentration in processing foods.
Cited references were well discussed.
This article would be improved if the author revised the following 3 revised suggestions as attached file.
1. Please chck the space between the text and cited reference number.
2. The cited reference, EFSA (2015) at page 9 line 378, has no reference number.
3. Please check reference format of several references. Some journal abbreviation have period but dome do not have.
I am not a native English speaker. The manuscript seems have no major mistakes are detected and the manuscript can be easily understood and read. The cited references are well discussed.
I enjoyed reading the review paper; the needs of special groups of cereal based food processing and baked foods. This manuscript presents useful formulation and processing strategies for food processor.
Date of this review
6 April 2023 21:48

Author Response
Dear Reviewer,
Thank you very much for your valuable opinions and comments to improve our paper. Corrections and additions were done in the text. Please find the point-by-point response below:
1-Please chck the space between the text and cited reference number.
AU : It was checked and corrected throughout the text.
2-The cited reference, EFSA (2015) at page 9 line 378, has no reference number.
AU : It was added.
Please check reference format of several references. Some journal abbreviation have period but dome do not have.
AU: Journal abbreviations were corrected.
Round 2
Reviewer 1 Report
Dear Authors, I still consider the manuscript unsuitable for publication in Int. J. Environ. Res. Public Health. The integrations and revisions do not substantially change the quality of the manuscript and do not bring significant novelty with respect to previously published papers (a review on the same subject was recently published: Rifai and Saleh, DOI: 10.1177/1091581820902405) . The only table presented in your work reports data from just one reference (EFSA); I recommend to see the paper by Koszucka et al. (https://doi.org/10.1080/1040839888222) as an example of literature data synthesis and presentation (tables 2 to 5).
Author Response
Dear Reviewer,
Thank you very much for your comments and paper recommendations. EFSA is European Food Safety Authority and EFSA reference is the report that includes numerous studies and results in the field. As the only one reference, it shows us the general situation. Thank you, I will take into consideration your recommendations. I respect your final decision.